# Dealing with Daily Rhythms: Families' Strategies to Tackle Chronic Time Pressure

**Guillaume Drevon** [1,*]**, Philippe Gerber** [2]  **and Vincent Kaufmann** [1,*]

[1]  Urban Sociology Lab, Swiss Federal Institute of Technology Lausanne (EPFL),
    CH-1015 Lausanne, Switzerland
[2]  Urban Development and Mobility Department, Luxembourg Institute of Socio-Economic Research (LISER),
    LU-4366 Esch-sur-Alzette, Luxembourg; philippe.gerber@liser.lu
[*]  Correspondence: guillaume.drevon@epfl.ch (G.D.); vincent.kaufmann@epfl.ch (V.K.)

**Abstract:** As suggested by the conservation of resources theory, in contemporary societies time is considered as a limited resource in the same way as money and energy. In the current paper, a novel daily rhythm approach related to motility is presented, in order to highlight the effects of life acceleration on family life management and other professional, leisure, and consumption activities. The analysis is based on a qualitative survey involving 20 families (40 interviewees) that include long-distance commuters living in the suburban areas of Voiron and Thionville in France. These families are composed of an active couple and at least two children under 18 years of age, and the couple commutes at least 60 km every day between home and work. Based on this particularly stressful daily configuration, the qualitative survey deals with the modalities of managing daily time between and within these couples. The quantitative and qualitative analysis of the corpus of interviews shows first, a very high daily rhythm, and second, the diversity of strategies that lead to a typology of resources used to deal with daily time pressures. The results suggest that forms of time-related vulnerabilities depend on social, economic, and temporal resources, while confirming the importance of rhythms analysis in the daily mobility field and in the resource theory.

**Keywords:** time pressure; motility; family; mobility; resources; couples; inequalities

---

## 1. Introduction

Like money and energy, time is a finite resource [1]. The feeling that this resource is insufficient is likely to generate stress phenomena at the individual level. Time pressures affect social life and tend to reinforce gender inequalities [2–6]. Social acceleration results in a constant feeling of a lack of time [7,8] and less satisfaction with daily life [9]. The mention of acceleration by Hartmut Rosa [10,11], suggests that individuals in contemporary Western societies experience increasing acceleration in the pace of life. This acceleration of life rhythms is a result of the combination of multiple types of restrictions to activity and mobility, together with an increase in the range and opportunities for mobility and daily activities [12].

Daily mobility is a symbolic example of the acceleration of contemporary societies and its harmful effects on individuals. Improved means of communication and transportation have increased the potential for crossing space, along with multiple demands for mobility and activity [13]. For example, improved accessibility tends to result in a greater distance between people's homes and workplaces [14,15]. This intensifying remoteness in Western countries in turn brings about increasingly long, tiring, and time-consuming commutes [16–20], which add to the list of other sources of time pressures related to the management of family life and work demands [21,22]. The temporal approach presented in the current paper seeks to put into perspective the implications of the association between

long commutes to work, and the significant time pressures related to work and the management of family life. As suggested both by Hartmut Rosa and in recent research, it seems interesting to focus on this association with daily time for people whose temporal resources remain scarce [4,23,24].

This study explores the connections between the spheres of work, family, and commuting. Specifically, it aims to test the main hypothesis that the reconciliation of time in daily life requires specific skills and resources. These skills and resources—which make it possible to link motility to spheres of activity—contribute to developing the ability to reconcile times and spaces in daily life. To test this hypothesis, we use a corpus of interviews conducted with 20 families. The interviewees were selected according to the profiles and criteria representative of dual-earner couples with children, where both adults commute to and from work daily [25] and where children are doing activities sometimes accompanied by parents. These couples were mostly asked about their representation of daily time and the ways in which daily life within the household is organized.

The approach we adopt regarding time reconciliation comprises considering daily life as a part of a constrained framework that offers a limited time budget for carrying out all the necessary activities. This study takes a cyclical approach that covers the three structuring elements of daily life and their interactions: the professional, family, and mobility spheres. The professional sphere concerns activities associated with work; the family sphere refers to household-management-related activities; and the mobility sphere concerns the location of professional and family activities, as well as the movements allowing access to them. Each of these spheres is associated with specific temporal obligations. The conjunction of the professional sphere and the family sphere leads to the confrontation and coordination of household members' time frames. The obligation to travel to work (interaction between the professional and mobility spheres) involves commuting journeys, which may be extensive. Lastly, the interaction between the family sphere and mobility sphere refers to the obligation to accompany young children to their places of activity. The combination of interactions between the spheres of daily life sets the terms of the spatio-temporal equation for family life [26]. These interactions reflect the interdependence of spheres of daily life.

The article is organized into three parts. Based on the current state of knowledge, the first part is dedicated to conceptual positioning. The second presents the survey methodology and data analysis. The third part focuses on the presentation and discussion of the main results.

## 2. Conceptual Framework: Stress, Motility, and Time Reconciliation

### 2.1. Household Stress and Time Pressures

Since the 1970s, research into daily mobility has been strongly influenced by time geography [27–29]. Time geography involves the simultaneous consideration of space and time to understand how daily activities and movements are enacted. These modalities are partly shaped by three types of constraints: physiological, social, and power constraints. To carry out their programs of activities, individuals must adapt to these constraints, while ultimately having limited temporal and spatial resources at their disposal.

Although this approach to mobility has helped lay the foundations for thinking about accessibility [30,31] and the modeling of activity programs [32], time geography does not concern the use of time [33] underlying the logic of resulting actions [34–36]. The choices in organizing daily mobility are nevertheless based on important trade-offs, which tend to reconcile individuals' spatio-temporal constraints and limited time budgets in the circadian cycle. These trade-offs are likely to generate different forms of temporal pressure and stress that tend to adversely affect people's quality of life. To complement the approaches developed by time geography, it is necessary to address the way daily life is conducted in constrained temporal contexts by examining the representations of time.

Understanding the perception of time in contemporary forms of mobility is an important issue, particularly in the general context of accelerating mobility rates driven by the development of remote communication systems and increased accessibility [37]. People have shifted from lifestyles in which

activities and roles followed one another over time, and where this succession generally involved travel, to lifestyles with "mixed" times, marked by the speed of succession and the multiplication of activities through communication over distance. Several recent studies have shown that the development of greater mobility reflects the meeting of rapid transport, remote communication systems, and their uses in a context of strong pressure to travel [38,39].

The increasing diversity of household members' activities requires important movement skills and the ability to reconcile time and space [35]. This is particularly important for households in sparsely populated areas that are poorly served by public transport. In cases where children are not yet autonomous in their daily journeys, a car is often indispensable. Each member of the household has his or her own daily and weekly schedule of activities, and mobility requirements are at times contradictory and involve trade-offs and negotiations within families [40]. In this context, households are confronted with a situation of spatio-temporal dissonance between members—dissonance that contributes to increased temporal pressures on the couple. Interactions between household members result in travel arrangements (accompaniment, carpooling, etc.) ensuring that each member can carry out his or her activities [41]. This situation directly influences the modal choice, which is partly determined by the interactions and arrangements [42].

Travel itself is also a source of stress, especially with regard to journeys to work. Long commutes by one household member are detrimental to the family sphere and may increase the risk of a couple separating [43,44]. Mobility contingencies also tend to play a role in increasing stress levels. Similarly, in line with work that demonstrates a link between travel time and stress levels [16,18], several authors have examined the relationship between these factors by comparing modes of transport [17] and working hours [45]. These studies suggest that travel-related stress is greater for motorists with fixed work schedules. Those who travel by train and have flexible working hours are less affected. Depending on household mobility patterns and the needs of the members, the mobility sphere can be a source of time pressure, especially if travel time is long.

## 2.2. Work and Family Spheres: A Temporal Confrontation

As the conservation of resources theory suggests, time is a finite resource, as are money and energy [1]. The feeling of this resource being insufficient is likely to generate stress at the individual level [3]. Accordingly, the feeling of not having enough time is an important marker of contemporary lifestyles [2,3,5]. Literature suggests that the temporal pressures to which individuals are subjected are mainly found in the relationship between the work and family spheres [46–48], and time pressure arises when individuals lack temporal resources in either of these. This pressure is all the more important when the family and professional spheres compete with each other, thus, involving decisions and negotiations about the allocation of activity time budgets. In this bilateral relationship, work can interfere with family and vice versa [49]. The resulting conflicts can lead to depressive syndromes and a lack of satisfaction with work performance and family involvement [50]. Literature shows that a large amount of time allocated to certain areas of life leads to conflicts with other areas: a phenomenon of structural imbalance in the arrangement of time in daily life [21]. Time scarcity can be a daily constant, and in the life of individuals in general [3], which is linked to an increase in the potential availability of activities and the demands for mobility, and in turn the activity it induces [51,52]. For some authors in the field of psychology, temporal pressure is the result of frustration that is rooted in the difference between the number of activities carried out by an individual and the potential number available in his or her environment [53]. Given such conditions, the effects of time pressure have been shown in numerous studies [54,55] and are often the prerogative of executives and higher social classes [56]. The research conducted on this subject mainly concerns working time, where time pressures are a result of the demand of professional productivity and the phenomenon of burn-out is the emblematic figure.

### 2.3. Skills for Reconciling the Times and Spaces in Daily Life

According to the definition by the Chicago School [57], mobility is "the intention and then the achievement of the intersection of geographical space involving social change." Using this definition, it is possible to focus the investigations on the possibilities of moving, then on mobility intentions, and then on the transformation of potential and aptitudes into actual movement. In this approach, each person or group is characterized by greater or lesser apparent abilities to move on a daily basis.

All these abilities can be conceptualized using the notion of motility [58]. At the individual level, motility is defined as "the set of characteristics specific to an actor that allows him or her to be mobile, i.e., physical abilities, income, aspirations to be sedentary or mobile, social conditions of access to existing technical transport and telecommunication systems, acquired knowledge, such as training, driving licence, international English for travelling, etc." [58]. Motility also refers to the social conditions of access (the conditions under which it is possible to make use of what is available in the broad sense), knowledge and skills (that are required to use what is available), and mobility projects (the actual use made of what is available). The interest in motility to describe and understand life rhythms lies in the fact that the reconciliation of activities in time and space refers to a set of core competences within motility to cope with the injunction to mobility that constitutes contemporary lifestyles. However, not all people seemingly have these competences in the same way. The ability to be mobile has become an essential resource, or even capital [58], for socio-professional integration, not only in terms of the quantity of motility but also in terms of its qualities. This is notably explained by Hanja Maksim [59], who highlights mobility cultures specific to social groups, particularly in terms of skills.

In addition to travel skills, research concerning working people shows that they are more likely to seek to control their time by adopting a reflective stance with regard to their agendas [60]. This reflexivity allows them to adopt temporal attitudes that facilitate balancing the different spheres of activity in social life. Several studies have identified different types of reflexivity—depending on the complexity of the temporal configurations—from which different attitudes are adopted by working people. These people attempt to reduce temporal pressures and stress in the workplace [61] and in their social life [62] by putting into perspective the importance of coordination [63], strategies [64], and downtime [22,65]. The literature on this topic focuses mainly on the field of work. Other research has explored strategies for managing daily time between work and family life spheres [4]. This research puts into perspective the various tactics used to reconcile time, space, and activity spheres in daily life. These tactics are based, for example, on the adaptation of work schedules [24], "multitasking" made possible by information and remote communication technologies [66], teleworking [67], and the organization of work and family life [4]. The strategies implemented to cope with daily rhythms also rely on a couple's negotiation skills for managing children [23] and on a network of family and friends [68].

### 2.4. Questions and Assumptions

The current article focuses on the time pressures experienced by couples with children, in order to highlight the skills developed within households coping with conducting daily life with varying degrees of satisfaction. More specifically, using the conceptual grid of motility the aim is to understand how families subjected to constant life rhythms reconcile their spaces, times, and spheres of life. From this perspective, the skills for reconciling daily spaces and times become tools for managing the three spheres (professional, family, and mobility) of daily life. Thus, the general question is: considering that families' pace of life is constant, do they have specific skills enabling them to manage their daily activities and the temporal pressures they face? This general question can be broken down into two working hypotheses. First, these aptitudes are based on resources internal and external to the families that have yet to be identified and characterized (H1). Second, adapting to the daily rhythm implies important concessions that tend to reduce time devoted to leisure (H2). The hypotheses were tested based on a corpus of 40 interviews conducted with 20 families in particularly tense mobility configurations.

## 3. Material and Methods

The second part of the article presents details of the survey and the methodology for analyzing the corpus of interviews.

### 3.1. Study Sites and Survey Populations

The survey was conducted between September and December 2015, and included a panel of 20 families. Therefore 20 couples (a total sample of 40 people) were interviewed. The families reside in the suburban agglomerations of Voiron (n = 10) and Thionville (n = 10) in France. Voiron and Thionville are included in the functional metropolitan areas of Grenoble in France and Luxembourg in the Grand Duchy of Luxembourg, respectively. These two study areas were selected to allow an international comparison between two study areas: Voiron–Grenoble and Thionville–Luxembourg (Figure 1). Although the Thionville–Luxembourg (study area 1) area is a borderland and the Voiron–Grenoble (study area 2) area is not, they still have important similarities in terms of spatial and social aspects. At the level of spatial configuration, the morphological agglomerations of Thionville and Voiron are located equidistant from the main poles of Luxembourg (27 km) and Grenoble (26 km). An urban discontinuity marks the separation between work and residence agglomerations in both cases. The two sites have similar structuring communication axes: a motorway, a departmental road, and a railway line connecting both the sending (Thionville and Voiron) and receiving (Luxembourg and Grenoble) centers, thereby offering good potential accessibility. In 2015, in terms of resident populations, 7500 working people lived in the Thionville and Voiron agglomerations and worked in Luxembourg and Grenoble, respectively. Preliminary analyses carried out on the basis of quantitative surveys of the two sites show that these populations have important similarities in terms of household composition, age, level of education, and socio-professional category. The majority of employees in Voiron and Thionville are couples with children. They have similar free time and forced time budgets. A comparative analysis of the spatio-temporal behavior of the two populations also suggests important similarities in the modalities of activity deployment (see details in [69–71]).

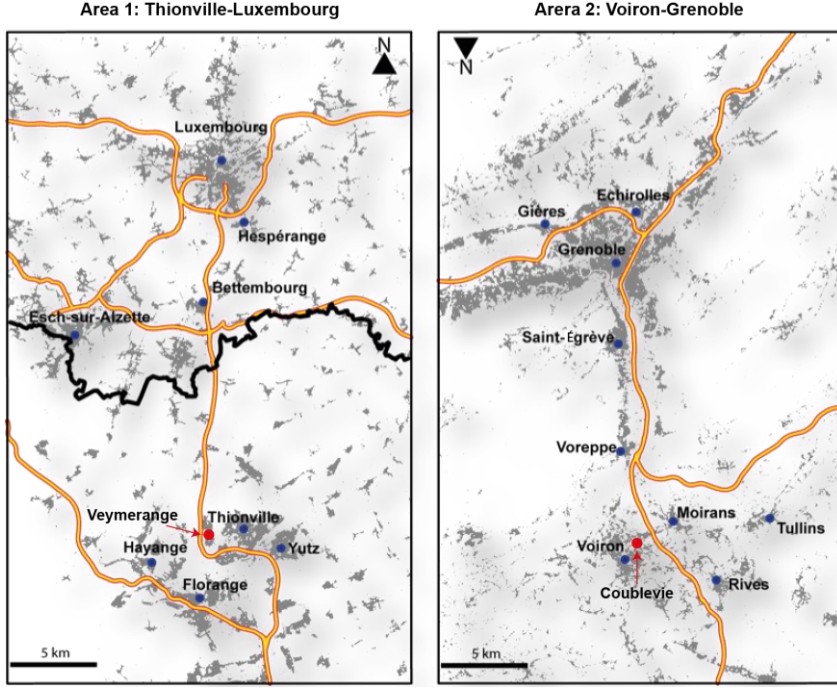

**Figure 1.** Geographical location of the two study areas.

To recruit households for the qualitative survey, the protocol relied on data from existing mobility surveys in the two areas: the Household Travel Survey of the Grenoble urban region

and the Average City Travel Survey of the Thionville urban community (more information is available at: https://www.cerema.fr/fr/activites/mobilite-transport/connaissance-modelisation-evaluation-mobilite/observation-analyse-mobilite/enquetes-mobilite-emc2). The data show the statistical sectors characterized by an over-representation of households that are potentially exposed to strong temporal pressures according to two socio-demographic criteria (i.e., dual-earner couples with one to three children under 18 years of age) and one spatial criterion (i.e., at least one of the partners works in the Luxembourg or the Grenoble agglomeration). Therefore, 10 families residing in the Thionville agglomeration and another 10 families in the Voiron agglomeration were recruited by randomly selecting addresses in the communal perimeters (Veymerange and Coublevie) of the morphological agglomerations of Voiron and Thionville (Table 1). The addresses collected were associated with telephone numbers, which were used to recruit the families that met the socio-demographic and spatial criteria mentioned above. The collection of telephone numbers was based on developing a sampling frame built from all the residential addresses in the communal areas of Veymerange (1408 addresses) and Coublevie (3036 addresses). All the addresses were then manually associated with telephone numbers taken from telephone directories. The sampling frame thus created made it possible to recruit households based on a random selection. The recruitment and investigation protocols were validated by the office of the National Commission for Information Technology and Liberties of the University of Grenoble Alpes (France). The protocol took into consideration the principles of respect for privacy at two levels (in accordance with the General Data Protection Regulation). First, recruitment zones were established on the basis of communal perimeters of residence. Then, the identification information for the households surveyed (surname, first name, address) was anonymized as the recruitment progressed.

**Table 1.** Description of the sample.

| Area 1: Thionville-Luxembourg | | | |
|---|---|---|---|
| Residence area | Households | Average age of the couples | Number of children |
| Veymerange | Family 1 | 37 | 2 |
| Veymerange | Family 2 | 38 | 2 |
| Veymerange | Family 3 | 43 | 3 |
| Veymerange | Family 4 | 44 | 2 |
| Veymerange | Family 5 | 39 | 3 |
| Veymerange | Family 6 | 50 | 2 |
| Veymerange | Family 7 | 43 | 3 |
| Veymerange | Family 8 | 42 | 3 |
| Veymerange | Family 9 | 43 | 1 |
| Veymerange | Family 10 | 44 | 2 |
| | | Average: 42.3 | Average: 2.3 |
| **Area 2: Voiron-Grenoble** | | | |
| Residence area | Households | Average age of the couples | Number of children |
| Coublevie | Family 1 | 37 | 3 |
| Coublevie | Family 2 | 42 | 3 |
| Coublevie | Family 3 | 43 | 1 |
| Coublevie | Family 4 | 40 | 3 |
| Coublevie | Family 5 | 38 | 3 |
| Coublevie | Family 6 | 44 | 2 |
| Coublevie | Family 7 | 42 | 3 |
| Coublevie | Family 8 | 45 | 2 |
| Coublevie | Family 9 | 39 | 3 |
| Coublevie | Family 10 | 42 | 3 |
| | | Average: 41.2 | Average: 2.6 |

Author: G.Drevon, 2019.

The recruitment and survey protocols were carried out in five stages. The first comprised the identification of households in the communes of Veymerange (Thionville–Luxembourg) and Coublevie (Voiron–Grenoble) that corresponded with the selection criteria. This step was based on randomly

drawing telephone numbers within these two communes. Households were accordingly contacted by telephone in order to determine if they met the three selection criteria mentioned above, based on a short recruitment questionnaire. To recruit the 20 families, approximately 600 households were contacted by telephone. The response rate was 42 percent (n = 252). Among these households, refusals to participate in the recruitment questionnaire were approximately 24 percent (n = 60). Of the 192 households that agreed to participate in the recruitment questionnaire, 23 percent (n = 44) met the selection criteria. The remaining 77 percent (n = 148) were single, active couples without children, or retired people.

In the second stage, a letter explaining the objectives of the survey (understanding the rhythms of life and the daily mobility of families) and the conditions of confidentiality (announcement, right to withdraw from the survey) was sent to the 44 households that met the selection criteria. Households were motivated to participate by being informed of the objectives of the survey, although the survey did not provide compensation.

In the third step, and following receipt of the letter, a second phone call was made to confirm participation in the survey and to arrange an appointment for a face-to-face interview at each couple's home.

The fourth step was comprised of interviews with the two members of each couple. As shown in Table 1, the households recruited in this way have important similarities. The average age of couples is 42.3 years in Veymerange and 41.2 years in Coublevie. On average, households in Veymerange have 2.3 children and those in Coublevie have 2.6 children. All the couples surveyed are dual-earners and in an intermediate professional position. It should be noted that couples composed of cross-border workers (area 1: Thionville–Luxembourg) have a higher income compared with the other two groups (area 2: Voiron–Grenoble). A higher income among cross-border workers is likely to make daily life easier, for example by allowing the use of home-help services (such as housekeeping or childminding).

*3.2. Methodology for the Survey and Analysis*

The investigation was based on semi-structured interviews [70]. From our research perspective, discourse was the preferred method for revealing social representations [71]. Interviews are an effective means of accessing social representations [72] and well-suited to the analysis of the meaning people give to their practices and representations. Here, the interview was used for comprehensive purposes. Semi-structured interviews allow us to deepen the knowledge of a domain or verify the evolution of a known phenomenon. By providing an important degree of freedom, the interviews offered the respondents an opportunity to express themselves according to their own logic. Three kinds of information were collected during the investigation. First, the families surveyed were asked about their representations of their daily conditions of mobility (e.g., Can you tell me about your daily traveling: mode, duration, satisfaction?) (the daily mobility sphere). The families were then asked about their representations of their rhythm of life in line with the three spheres of daily life mentioned above: satisfaction and representation of the daily rhythm of life at work and in the household (e.g., How would you describe the rhythm of your week in general (work, family, and household management)?) (the interaction between work and family spheres). Based on the conditions of daily mobility and more broadly daily life rhythms, the third part of the interview focused on the interviewees' abilities to reconcile daily times and spaces, through the different types of resources mobilized to cope with conducting daily life. In this part of the interview, the couples were asked about their mobility strategies (e.g., How do you manage your long commutes?), the distribution of tasks within the couple (e.g., How would you describe the distribution of tasks between you?) and negotiation between the partners (e.g., Do you have agreements between you for the organization of daily life?).

During the face-to-face interviews, both partners were present. This could imply a lessening of the discourse, especially in view of potential gendered attitudes and situations of domination. However, for the interviews as a whole, the speaking time was relatively balanced between the partners in each

couple. Depending on the time given by the couples for the survey, the interviews lasted between 60 and 90 min.

The analysis of the interviews was based on two methods. The first was quantitative, using occurrences in the respondents' discourse. The analyses were carried out on the basis of the structure of the interview and the three main themes: conditions of daily mobility, description of daily rhythms, and organization of daily life. Each part of the interview was analyzed using Iramuteq software. The frequency of the expressions and words used by the interviewed couples revealed the different lexical fields of the discourse, and this was subsequently deepened by the qualitative analysis. The second method, qualitative, focused on the content and meaning of the expressions used by individuals, based on an in-depth re-reading of the corpus and the lexical fields of interviews. The aim was to provide enlightening examples of the couples through their accounts of daily life, and to construct a tool for assessing their representations [73]. In this way, their representations can be approached using narrative methods, in order to work on the individual meanings and subjectivities of everyday life.

These two approaches are complementary. On the one hand, the analysis of occurrences measures the importance of the words and expressions used by the respondents [74]. On the other hand, it is important to understand and analyze in detail the meaning of individuals' discourses through an in-depth analysis of their strategies and verbatims.

## 4. Results

The results of the study are divided into three sections. The first covers aspects related to the sphere of daily mobility, which for all the respondents reflects many problems. The second section deals with constant daily rhythms and their representations. The third concerns the skills that families have developed to manage these rhythms better.

### 4.1. Major Difficulties in Daily Mobility

The occurrence analysis was conducted on daily mobility (Figure 2) and measured the frequency of use of each of the words associated with the topic (the first theme in the interviews). The results were classified according to five lexical fields that correspond to the sub-themes of the interview grid. The first field corresponds to the daily and weekly time frames, and mainly refers to daily times and the periods before and after work (mornings and evenings). Time frames (hours) were an important part of the discourse and include departure and return times to and from home, as well as dropping off and picking up children. Durations (minutes) also indicate the importance of travel time, which was frequently cited by the interviewees. Respondents instinctively referred to other periods (weekends) and days of the week (e.g., Wednesdays). The second lexical field is associated with transportation infrastructure (roads, etc.). The third lexical field concerns the difficulties encountered by people during their daily journeys. These are mainly comprised of problems related to congestion and traffic, and sometimes to hazards (weather, accidents, etc.). Modes of transport, the fourth lexical field, appeared as important elements of the discourse. The car was logically the main term used by people. The other transport modes (bus and train) were mainly compared with car use in terms of advantages and disadvantages. The fifth lexical field refers to the management of children, specifically in terms of dropping them off and picking them up in relation to school hours.

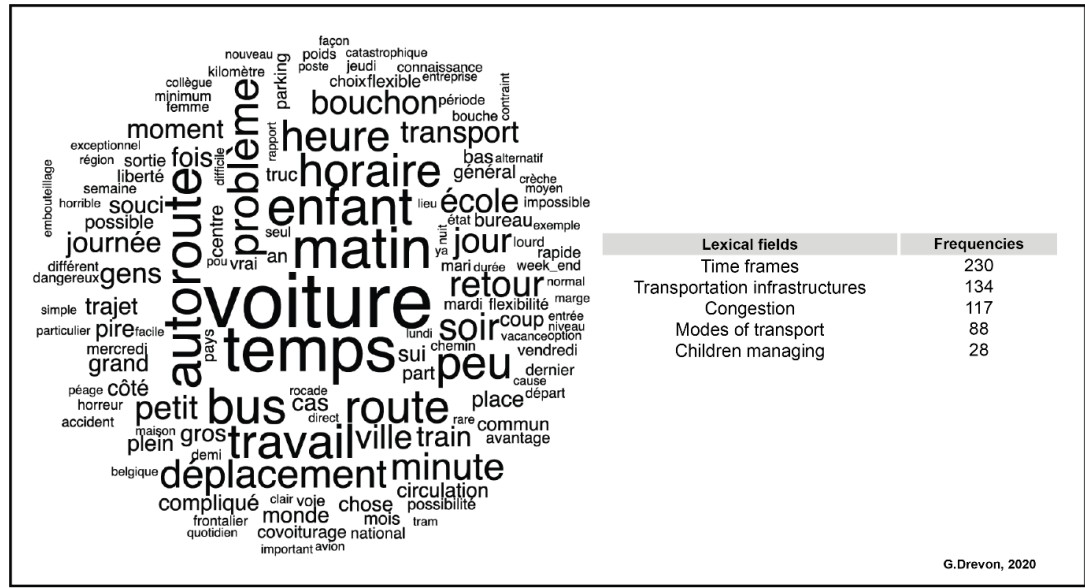

The table in the figure:

| Lexical fields | Frequencies |
| --- | --- |
| Time frames | 230 |
| Transportation infrastructures | 134 |
| Congestion | 117 |
| Modes of transport | 88 |
| Children managing | 28 |

G.Drevon, 2020

**Figure 2.** Results of the analysis of word occurrences about car users' daily journeys.

The analysis of occurrences shows that the discourse about daily mobility was organized around three main lexical fields: the times that correspond to the temporal boundaries of the day, transportation infrastructure, and transport modes. The latter two fields concern travel itself. The entire experience was judged negatively, reflecting the difficult conditions of daily mobility.

Commuting to and from work was most often associated with drudgery and described as slow (see transcripts in Figure 3). Home-to-work commuting time was considered a disadvantage, with a significant impact on the time usage of the day. Some respondents said that the time spent in the car generated fatigue and stress. This incompressible time also appears to complicate daily life and has repercussions at the household level. The results presented in this section corroborate the current state of knowledge on several levels. The literature and numerous reports show that commuting to and from work generates stress and fatigue that degrade the living conditions of individuals [16,17,75].

"That's the big disadvantage of our life here, for me it is. I spend way too much time here. In the car, it takes me two hours a day, one hour there and one hour back, and then it's very tiring. It's still annoying. It's complicated". Area 1, male

"The conditions are difficult because there are already so many of us. In winter, it's even worse in terms of travel. Traffic jams and accidents happen all the time, and police roadblocks don't help". Area 2, woman

"I leave in the morning between 7:30 and 8:30 and it's clear that access to Grenoble is complicated". Area 2, man

" Travelling conditions themselves are horrible. They don't make you want to go and work in Luxembourg. Some time ago, it was the apotheosis with the attacks, I used to travel for four hours a day". Area 2, woman

**Figure 3.** Examples of verbatims reports about commuting.

### 4.2. Constant Daily Rhythms

Analyzing the occurrences in discourses about the rhythm of life concerning daily activity programs (Figure 4) allows us to identify five lexical fields. Time frames (n = 493) are the main theme in these discourses, referring mainly to time boundaries and periods of the day. Words and expressions associated with managing children are the most frequently used, and indicate managing children to be

the most important concern (n = 139). Expressions relating to daily and regular activities rank third in terms of frequency (n = 100). The family also appears as a structuring element of the discourse (n = 50). Lastly, difficulties in carrying on with daily life are apparent in the discourses of the interviewees (n = 26). This first level of descriptive analysis reveals three important elements. First, the pace of life is mainly associated with time management and schedules on a daily and weekly basis. Individuals' discourses, generally centered on the family, show the importance of children and their activities in carrying out daily life.

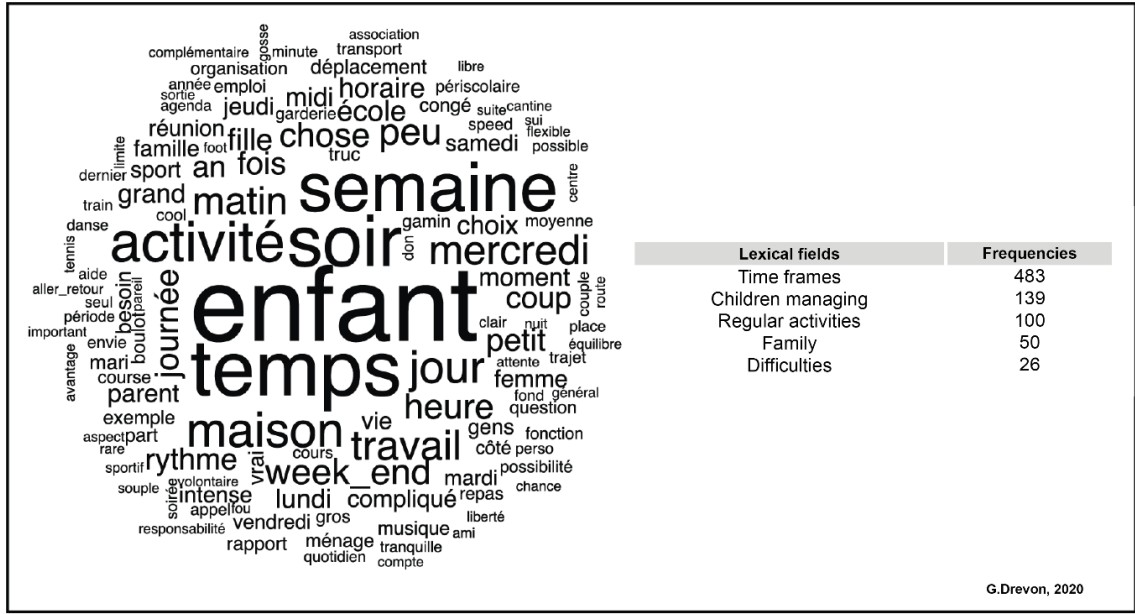

**Figure 4.** Analysis of word occurrences associated with daily activity rhythm.

For the majority of the people interviewed during the survey, the pace imposed by the daily work and household management activity schedules is considered constant (Figure 5). This is most often associated with the management of children. Children's time frames and the need to drop them off and pick them up in constrained time frames appear to be central.

Accordingly, particular times and time limits have to be respected by parents. Supporting children in their leisure activities is a structuring element in the planning of daily activities. The management of children and their mobility seem to weigh on the parents, as each child is involved in one or more activities outside of school time during the week. Work responsibilities and travel time also appear in the respondents' discourses, although little emphasis is placed on them. In fact, overall, the rhythm considered to be constant refers mainly to the children's activity schedules, which oblige parents to link the need to accompany their children throughout the day to work, daily travel, and household management.

An analysis of the occurrences of words and expressions associated with the pace of life concerning household management revealed five major lexical fields (Figure 6). As in previous analyses, time frames play a dominant role in the respondents' discourses. They mainly refer to the periods of the day and week when household management tasks are carried out (evenings, mornings, and weekends) (n = 228). There are tasks primarily related to everyday purchases (food and supplies) (n = 175). Then, care associated with children (meals, washing, and homework) is a central element in the discourses (n = 105). Lastly, the designation of household members (n = 76) echoes the fifth lexical field reflecting the distribution and organization of tasks (n = 57).

"The rhythm is monstrous, but then, again, we chose it." Area 1, man

"Unrestrained, one chose it, one does not complain about it. I prefer to move rather than remain doing nothing." Area 1, woman

"Go for the balloons..." Area 1, male

"I'd say it's a good job. It's running well, but there mustn't be too much sand thrown in the machine." Area 2, woman

"It's crazy. We have friends who say when they have a really shitty week, they say 'it's our family week'. We're the benchmark for the life of a douchebag, you know, a crazy person." Area 2, woman

"Well, four kids, too. For example, every night at least one kid comes home after 8:30 because he's got basketball. If it's not him, it's me who comes home late; after Thursday, it's you who's going to come home late too." Area 2, man.

"Yes, the pace is steady, that is to say there are the children's activities in the evening, and the leaving time is important. Sometimes, you have to cross paths. While one [child] has an activity, you have to make the others eat." Area 2, man

"It's still pretty fast, and as far as the kids' activities are concerned, there's no way we'd come back on purpose to pick one up and take it to a singing class or whatever." Area 2, man

"It's been a pretty tense week all around. Already, the kids mean less time at work. I can't work overtime that easily." Area 2, man

"When you add up the kids, homework, and activities, you can say that we have a really fast pace of life during the week and not a lot of time." Area 1, male

"We work our schedules around the kids' schedules." Area 1, female.

**Figure 5.** Examples of statements about the rhythm of activity schedules.

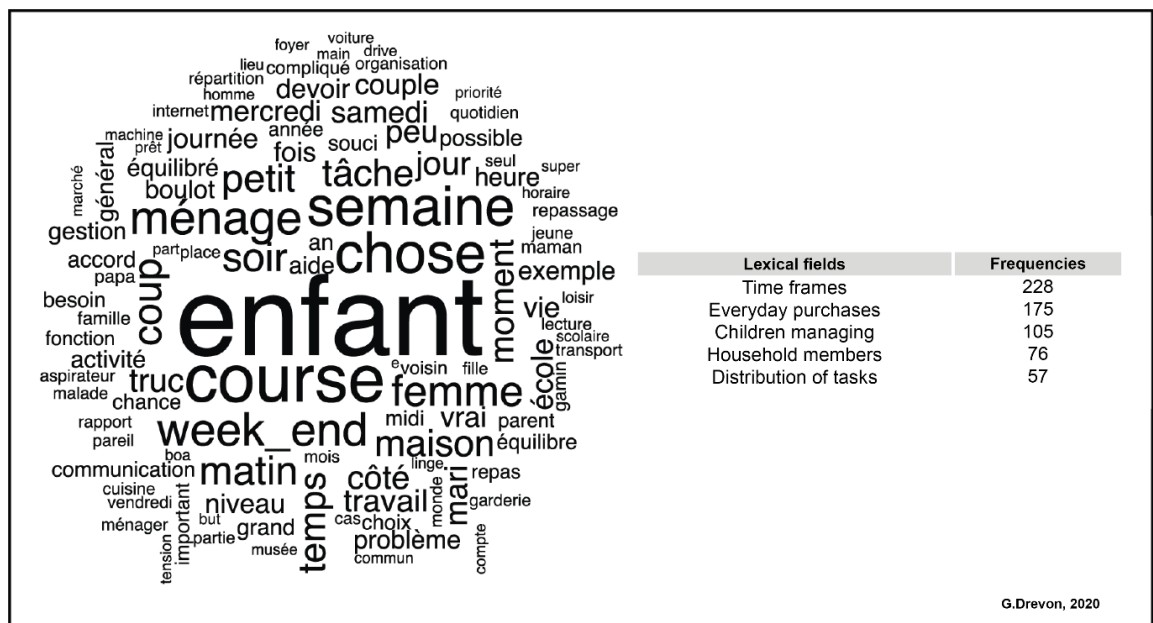

| Lexical fields | Frequencies |
| --- | --- |
| Time frames | 228 |
| Everyday purchases | 175 |
| Children managing | 105 |
| Household members | 76 |
| Distribution of tasks | 57 |

G.Drevon, 2020

**Figure 6.** Analysis of word occurrences associated with household management.

The survey reveals that couples feel they have little personal or marital time (Figure 7), as managing the children and the home form an important part of the day's schedule. These tasks put a great deal of pressure on the couples, who expressed frustration about the limited time available to them. The period after 8 p.m. seems the time that is more devoted to the couple together. Some of the interviewees tend to make certain periods of the week more "holy" or to leave their weekends as free as possible from the burden of the children. The couples' time appears to be a particularly important element in the life of the household in order to avoid desynchronization, which can lead to conflict

situations and even cases of separation. This situation also prompted the interviewees to encourage the coordination of time frames when tensions arise.

"You have to manage activities. What puts pressure on us is the whole thing. We don't actually live together." Area 1, man

"We have very little time for ourselves. We don't spend time together." Area 2, woman

"Around 9:30, it's not the right response, but we turn on the TV and we're a bit wrecked." Area 1, man

"Let's say we only have time for ourselves after 8:00 when the kids are in bed." Area 1, man

"After that, we don't have much time for each other. We try to go out for dinner, but it's rare. It happens, but it's rare." Area 2, man

"Then we have couple time after 9:00, actually, and then Saturday morning, which is in place now, because the kids are older, too." Area 2, woman

"Some weekends, we just get together, just the two of us. When my husband's here, he's really here." Area 2, woman

"When I'm working nights, we run into each other. I know couples who got divorced because of it. It got too complicated for them." Area 1, woman

"We cross paths quite a bit. There are times when we cross paths completely, but now we've taken our sides. We're trying to keep the places where we meet. We manage to coordinate, we had times when we lived next to each other… and it ended up in separation." Area 2, woman

**Figure 7.** Examples of verbatims about representations of the rhythm associated with household management.

Our analysis confirms the hypothesis regarding constant daily rhythms among dual-income households with children. This understanding also highlights important elements based on the relationship between rhythms and daily temporalities. To the best of our knowledge, few studies have examined this temporal dimension [12]. Nevertheless, it allows us to better understand the structuring elements of families' daily life from a more global vision, with the intent to summarize the interactions between the spheres of daily life. The analysis places three elements into perspective. First, couples are more concerned about the sequence of activities and the search for an appropriate agreement. Second, work appears to be a secondary element. Third, children are the central element in the daily life of the respondents, and the origin of important temporal pressures.

### 4.3. Daily Rhythm Management Skills

The analysis of the interviews conducted with border and non-border couples reveals three types of skills necessary for conducting daily life and managing the constant rhythms of life. The couples developed these skills through representations in the various spheres of daily life. These skills enable them to meet each other's needs in the household and reconcile the times and spaces of daily life.

#### 4.3.1. Ability to Avoid Saturated Space and Time

In daily travel, the ability to avoid congested space and time refers to the search for alternative routes (Figure 8). To avoid potentially congested areas of the road network, the respondents use their knowledge of secondary roads to take detours. This ability provides comfort and reduces the stress associated with commuting to and from work. Avoiding traffic jams by using secondary roads also appears to reduce uncertainty about travel time, which is closely related to the search for reliability and quality regarding travel [76]. Indeed, some respondents seem to favor stability with regard to travel time: choosing to spend more time on travel in order to improve reliability. Although this choice increases the travel time budget, it controls temporal hazards. Knowledge of secondary road networks is an asset that enables people to plan alternative routes. This aptitude demonstrates an initial capacity to adapt to the constraints of daily mobility. Avoidance skills can also be based on time management.

Faced with the constraints of daily travel, such as traffic jams, some of the respondents changed their departure times from home. Therefore, by leaving earlier or later, respondents felt that they benefited from better traffic conditions. However, this was only possible under certain conditions. In fact, most respondents have some flexibility in their working hours. In the case of those who work fixed hours, the ability to avoid time was more difficult to deploy.

"As time went on, we got to know the back roads." Area 1, male

"I never take the motorway, I take the national highway, the advantage of being a border resident and being here is to have Luxembourg, but it is also to know alternative routes. In fact, I only take alternative routes. I never take the motorway. It's less stressful." Area 1, male

"We now know all the back roads—the back roads and how to avoid traffic." Area 1, woman

"It's true that recently it's been rather catastrophic. We use the back roads, even in the evenings. There it is clear, Dudelange Volmérange or Kanfen, and after Kanfen, we go back to the back roads and the N-road to Saint-Michel and back to Thionville." Area 1, female

"I use secondary roads—national roads and smaller roads, where I drive a good fifteen kilometres more, it takes a good hour, but I drive in a more relaxed way because I avoid traffic jams even though driving on the national and departmental roads often requires more vigilance. I arrive in a less stressed state of mind at work, that's it." Area 1, man

"Small roads are indeed dangerous. It doesn't save us time, but it doesn't waste time, so we take them." Area 1, man

**Figure 8.** Examples of discussions concerning spatial avoidance skills.

### 4.3.2. Communication and Negotiation Skills Within the Couples

The discussion above indicates that dropping off children guides modal choice and sets the time frame for the home-to-work trip (Figure 9). Faced with this constraint, agreements between partners are put in place. The negotiations within couples—or the agreements—are aimed mainly at encouraging parents' temporal circumvention skills. Therefore, respondents tend to adapt their working hours to better manage the constrained framework that is set by the need to drop off and pick up children. One parent drops off the children in the morning and the second parent picks them up later. This organization makes it possible to shift the departure and return times for the first member of the couple and then for the second to get around rush hours and the difficulties associated with traffic jams.

"The fact that I postponed our work day with my husband... I asked never to work before 9:00 a.m. I'm the one who takes care of the kids in the morning until school starts. My husband leaves early in the morning and I come home later in the evening. I take the two children to school. My husband picks up the two big ones and I pick up the little one. The fact that he gets the two older children without the younger one allows the boy to do his homework quietly. Then I just have to pick up the little one." Area 1, female

"I'm a little late in the morning so I leave here relatively early. I leave at 6:30 a.m., so I'll try to get home between 6:00 and 6:30 p.m. so I can take care of the kids in the evening." Area 2, man

"Normally, I'm out of time. In the morning my wife leaves early, and I take care of the children. In the evening, it's the opposite. In the morning, I take the kids to school." Area 2, man

"One out of two times, one of us takes the little one to the nursery and the other to the school in Thionville. At the same time, the other one of us takes the child to school." Area 1, male

**Figure 9.** Examples of verbatims concerning spatial avoidance skills.

Communication and negotiation skills, particularly in relation to sharing the burden of care for children, are central to conducting daily life. Given the constant pace of life, the couples we interviewed insisted on the quality of communication (Figure 10), which appears to be an important element in the coordination and implementation of daily organization. Some time slots are dedicated to this exercise, which makes it possible to arrange the distribution of tasks and activities and report difficulties and dysfunctions.

"We communicate well. It's essential; otherwise, we can't function." Area 2, man

"Communication is how it's learned. When it's good, you have to talk; when it's bad, you have to talk too." Area 2, woman

"We have to be organised and communicate, and we manage to be complementary." Area 2, man

"We try to balance. Often on Saturday mornings, we discuss what we're going to do on the weekend. We organise ourselves like that. We try to communicate well." Area 1, female

**Figure 10.** Examples of verbatims concerning communication modalities.

The coordination, established through communication, reveals tight organization (Figure 11). Routines emerge that have been proven effective over the years, and seem to facilitate carrying out daily life. Each partner devotes himself or herself to a task within a specific time frame (morning, evening, etc.), which is reinforced by a relay system. The discourses of the people interviewed show a quasi-martial dimension, which refers to organization that runs with military precision within a precise temporal framework. Although this organizational capacity is an asset, it has the disadvantage of leaving limited space for maneuver when it comes to managing hazards.

"We've been doing it for 15 years, so that's the 'military' side of it; I totally agree. We have to get organized and communicate to complement each other." Area 1, man

"We try to organize ourselves, that is to say, do as much as we can on weekends and also on Wednesdays. If we do it on a daily basis, it's not possible, so there again, it's like a military organization. It's the same." Area 1, man

"The week is military like. We don't have a choice. We have to be organized and the kids do things quickly and do something left and right. If we don't take care of it, they're still dressed and doing something at 9 p.m. It's 'military'. You go home and do this and that. He does it in the morning and I do it in the evening." Area 1, woman

"We try to communicate well. We've chosen a certain rhythm. It's often time-measured, but that leaves a bit of room for spontaneity from time to time." Area 1, woman

'Time off work is pretty tight. Overall, you have to be well organised'. Act 1, man

**Figure 11.** Examples of verbatims concerning daily organization.

### 4.3.3. Social and Support Network Skills

The family or neighborhood networks (Figure 12), which is made up of people from outside the household, seem to occupy an important place in carrying out daily life, and more particularly in the management of children's activity schedules. Social skills and the ability to make use of a support network enable the partners to mobilize one or more people from outside the household to help carry out the planned activities. According to the discourses of the interviewees, this is mainly a question of picking up children from school or an activity. This practice can have a permanent temporal dimension and be part of the couple's routines outside the home. Therefore, it helps to relieve temporal pressure on individuals by exempting them from a constraining activity, such as picking up the children from school.

> "Now, when I'm all alone, it's the same. We have the neighbor across the street, where I drop the kids off. She drops them off at school, so they don't have to go to daycare in the morning and evening." Area 2, woman
>
> "The thing is, we come from the Paris region where it's three times worse. So, overall, we think it's shitty, but it isn't so bad. The neighbors help us out and we don't have to be completely rushed; otherwise, we'd be really rushed." Area 2, man
>
> "So, on Monday, we take the kids to school. My in-laws pick them up and wait for us before eating. They wait for us in the evening before we go home. So, we have flexibility." Area 1, female
>
> "I have my parents who live nearby. So, my parents come to pick them up after school." Area 1, female
>
> "We've set up a new thing that we do every other week, with neighbors who live nearby, who have children in primary or secondary school." Area 2, woman

**Figure 12.** Examples of verbatims concerning solidarity.

The forms of solidarity that emerge more than once during unplanned events are mainly related to the management of contingencies (delays, etc.) (Figure 13). In fact, if a couple is unable to carry out a program of activities—for example, those involving the children—they can call on an outside person (neighbor, parent, etc.) who can take charge of the activity planned beforehand to avoid a problem. Regular solidarity contributes to the sustainability of the daily rhythm. However, as soon as a hazard disrupts the routine, the situation seems to become more complex (Figure 12). The problem of the "sand thrown in the machine" appears to be an anxiety-producing element for couples confronted with an unplanned situation. In concrete terms, the solidarity observed in these cases is based on either one partner in the couple or an outside person from the couple's social network.

> "So, now [the bus was late], it's 'operation phone.' 'It's the neighbours, can you help me out?'" Area 2, woman
>
> "At the beginning of the week, my wife was in Paris and I was alone, so I called on friends and mothers of friends, and some of them helped me take them to the activities and so it's sorted out. But it's true that it's still quite a rush." Area 1, male
>
> "In fact it turns out well when there are no grains of sand seizing up the machine. But, typically, just the blow of being called by the school because a child has fallen, when he is sick. If it happens during the week, we have to find a way to deal with it. Often, it's the phone call, 'can you be free today?'" Area 2, man
>
> "Basically, I'd say it's a very good job. It's running well, but there shouldn't be too much sand in the machine. As soon as it starts to happen, as soon as something unexpected takes place, we manage it. The bigger they are, the better it is managed. But when we get calls in the middle of the afternoon, 'your daughter, she has fallen,' we have to come." Area 2, woman
>
> "I'd say we're doing relatively well. It's true that it happened twice in those four or five years that the kids were left high and dry. But it was really related to the snow. The ice blocked everything and then we had to call on friends or neighbours to get them back, but that's extremely rare." Area 1, male

**Figure 13.** Examples of verbatims reports on contingency management skills.

A significant proportion of the interviewed couples use outside services to help manage the household. Beyond the drudgery associated with managing household chores, the couples are more interested in securing free time, which is then reinvested in leisure activities or allows them to devote more time to their children. This strategy seems to reflect a concerted decision in the face of the time pressures associated with managing the household, which requires adaptation. The decision to employ someone to relieve the couple of household management tasks also relates to the economic resources that make it possible to acquire more free time.

## 5. Discussion and Conclusions

The survey of 20 dual-earner couples with children highlights the time pressures faced by families. These pressures were expressed in different intensities. Daily mobility and (particularly) the combination of long commutes to and from work and support for children appear to be the main sources of time pressure. Household management is also an important source of temporal pressure, which is made apparent by the relatively limited free time budget for the couples. Lastly, time pressures related to work demands were rarely mentioned in the interviews; couples generally tended to mention the difficulties encountered in daily mobility and managing family life. This reflects a commitment to regimens [77–81] that are mainly centered on the family and on carrying out daily life within the household [35]. On the whole, the survey suggests that families are subject to a constant daily rhythm of life, in which time management is a central issue [21,47]. The harmful effects of this rhythm of life are reflected in a chronic lack of time and feelings of stress and fatigue. Our results tend to confirm previous work, taking a critical approach to the transformation in contemporary societies [2,3,5,10] of the relationship between time and its harmful effects on individuals. These prior studies put into perspective the role of life course stages in the fluctuation of daily rhythms, which depend on the family configuration and residential location.

The observation of the centrality of time management in the context of time pressures clearly reveals the importance of motility as a key resource for balancing work and family life. Organizing the daily life of a family requires excellent knowledge and mastery of each family member's ability to move, as well as the management of their respective mobility in order to ensure a daily life that is workable in terms of life rhythms.

The results also testify to the reflective processes that lead to changes in attitudes about conducting daily life [60]. The skills we identified were expressed through the practices that had been developed in the three spheres of daily life and are skills based on cognitive, temporal, communication, economic, and social resources. Putting these different types of resources into perspective confirms the first research hypothesis (H1), as the resources enable couples to implement several strategies. The temporal pressures related to daily mobility are the subject of travel skills that tend to reconcile the constraints of social interaction and travel time budget. They materialize through the avoidance of spaces (congestion zones) and times (peak periods) that are likely to generate travel difficulties. With regard to the temporal constraints associated with children, the management of daily mobility also relies on communication and negotiation skills between partners. In practice, this ability results in the distribution of mobility loads in accordance with the couples' spatial and temporal constraints. The skills developed by couples in the face of daily mobility constraints are based on different types of resources. The first is a detailed knowledge of transport networks (cognitive resource) and the possibility of benefitting from flexible working hours (temporal resource). Second, the potential for communication and negotiation (communication resources) between partners enables the organizing of mobility management, which is apparent in the daily routines [82–85].

The management of children, and family life more broadly, is also a source of significant time pressures in daily life. In this area, couples rely significantly on their social skills, which materialize through building a social network of family and friendship solidarity and the ability to mobilize people who are likely to be in a position to provide support for conducting daily life. In practice, these social skills translate into soliciting family members and friends, mainly to facilitate children's activities. This skill also plays a prominent role in the management of risks, particularly those related to difficulties in daily mobility. This second skill is particularly associated with the extent, location, and density of the couple's support network (social resource). The social resource and the associated potential support constitute a major stake in carrying out daily life. It also refers to the intervention of the couples' parents in the management of family life [85,86]. Managing the home and the tasks associated with it is also a major issue for the respondents. To cope with these demanding tasks, they rely on their communication and negotiation skills to divide the load between them. They also use their ability to employ the services of external providers, who are mainly involved in home maintenance

and childcare [87]. These paid services use the couple's economic resources. Therefore, inequalities are likely to appear between families, depending on their income level.

Generally speaking, the couples interviewed are subject to substantial time pressures, the management of which requires the implementation of complex strategies that rely on a variety of resources. In this configuration, time pressures leave little time for leisure activities or time for the couple, thus, confirming our second hypothesis (H2).

However, some limitations of this research should be mentioned. Even though the qualitative survey was designed based on analyzing quantitative surveys carried out at the two sites, the small number of households interviewed limits any generalization of the results. Furthermore, both partners from each couple were interviewed at the same time. This interview configuration could lead to concealing gendered dominance relationships. Lastly, the couples interviewed are mainly middle class or financially well off, thus benefiting from significant economic resources for managing time pressures. It would have been interesting to compare these families with households that are less well-endowed economically and/or in a single-parent situation, in order to put into perspective, the inequalities in the management of daily rhythms. Car use is particularly important in the families interviewed. These families live in areas where access to public transport is low. In order to put the role of different modes of transport in managing daily rhythms into perspective, it would have been relevant to compare families living in different spatial contexts. In future research, a comparison between families living in suburban and urban areas could show the differences in terms of daily rhythms and the use of different modes of transport.

The empirical investigations presented in this article contribute to a better understanding of the skills needed to manage constant daily rhythms. The skills identified in the framework of the survey support previous work on motility [58], and, more broadly, the ability to reconcile living space and time [12]. In addition to motility skills, the identification of resources [88] enables us to put into perspective the conditions necessary for couples to develop their skills to manage daily life. More specifically, the research presented in this article allows on the one hand reinforcing the concept of motility by showing the importance of mobility skills for conducting daily life. On the other hand, the research allows us to also reinforce the concept of motility on two levels. First, time management skills appear to be a major element of motility. Second, social skills relating to coordination and the ability to use social resources in daily life also contribute to the concept of motility. In terms of mobility, and more particularly in terms of modal choice, the research results show the importance of coordination between household members in the choice of transport mode [89,90]. Indeed, although the analyses did not focus on this aspect, all the activities concerning children are carried out by car, thus implying an important socialization of the children to car usage, particularly in low-density areas [91]. Accordingly, inequalities between families are likely to appear in this research. These may be apparent through a lack of mobility skills, modest income, or social isolation. Moreover, situations of domination—for example, within a couple—may contribute to imbalances in the distribution of activities and time constraints.

In a more general context of accelerating life rhythms, this research prefigures important research prospects. At the first level, it is a question of developing an understanding of daily rhythms that appear to be a central element of well-being in contemporary societies. At the second level, the resources necessary for self-maintenance [92] in an accelerating society are likely to reinforce class and gender inequalities [93].

**Author Contributions:** Conceptualization, G.D., V.K., and P.G.; methodology, G.D. and P.G.; software, G.D.; validation, G.D., V.K., and P.G.; formal analysis, G.D.; investigation, G.D.; resources, G.D.; data curation, G.D.; writing—original draft preparation, G.D.; writing—review and editing, V.K. and P.G.; visualization, G.D.; supervision, P.G.; project administration, P.G.; funding acquisition, G.D. and P.G. All authors have read and agreed to the published version of the manuscript.

**Funding:** This research was funded by the Luxembourg National Research Fund: 1395509.

**Conflicts of Interest:** The authors declare no conflict of interest.

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
