# Peer review of "Dealing with Daily Rhythms: Families’ Strategies to Tackle Chronic Time Pressure"

_sustainability, doi:10.3390/su12177193_

Round 1

Reviewer 1 Report

sustainability-835158

Dealing with daily rhythms. Family’s strategies to tackle chronic time pressure

The aim of this paper was to understand how families subjected to constant life rhythms reconcile their spaces, times, and  spheres of life using the conceptual grid of motility.

This paper addresses a interesting and common topic in scientific literature with important practical implications. The objective is relevant and the methodology is adequate. There are, however, some aspects that should be reviewed. In my opinion, would be necessary to resolve some doubts regarding the method and to justify and expand some aspects of the discussion and conclusions.

 RECRUITMENT

Could the authors provide more information about how they recruited the participants? Were there participants who did not agree to participate when contacted?Did participants get any compensation?What information were participants given about the aims of the study when they were encouraged to participate? Was it necessary for the study to be approved by an Ethical Committee? 

PARTICIPANTS

I think more information about the participants is needed. Could the authors provide more information about the sociodemographic and professional variables of the participants? The authors say that these people share similar socio-demographic characteristics, but which are these variables? For example: employment, age of children, income, and educational level, family living in the same neighborhood, suffering from illness (parents or children). Controlling these variables is important because it can affect the results 

INTERVIEWS

-     It would be convenient to provide more information about the content of the interview: indicate the structure and examples of items, order in which the themes were presented and indicate whether the authors relied to design it in a previous interview.

-  It would be convenient to include the approximate duration of the interviews and the maximum and minimum duration.

- The authors say participants are recruited by phone, but it is unclear how the interviews are conducted. Are interviews also conducted by phone? Is it always the same interviewer?

-  The timing of the interview is also unclear. Is a first contact made to explain the objectives of the study and then a day is arranged to carry out the interview?

- It is indicated that both members of the couple participate. Is one member present while the other answers? That could affect the results.

RESULTS

I find the two approaches (qualitative and quantitative) interesting, but I have doubts about how they were carried out.

- Quantitative analysis: Did the authors establish prior criteria on which words or expressions were to enter the analysis of occurrences? What were the sub-themes of the interview? Was there a previous synonym analysis? Only verbs and nouns related to the topic were included?  

- Qualitative analysis: How many researchers participated in this analysis? Was an agreement between judges necessary? Was some kind of program like Atlas-ti used? 

DISCUSIÓN

The discussion is well argued, but the authors should deepen further into the implications of the results. The relationship between hypotheses and discussion is also not clearly seen.Do the authors think that the strategies expressed by parents are appropriate? They must deepen more into the solutions or skills that couples must have or about the kind of strategies to improve those skills or resources.

The limitations of the study must be included.

Finally, the authors must clearly indicate what the study contributes to the literature that already exists.

In summary, I think the study is interesting, but it is necessary to provide more information on the issues outlined above.

Author Response

We would like to thank reviewers for their highly valuable comments. The reviews were particularly beneficial for the improvement of the article, especially in terms of clarifying the methodology and the contributions in relation to the current state of knowledge.

Our responses to the reviews are formally presented as follows: referee comments in black and responses in blue. In addition, the main changes have been reported in green font in the article.

As a general comment, please note that the language has been improved with the help of a professional English proofreader.

Reviewer 1

The aim of this paper was to understand how families subjected to constant life rhythms reconcile their spaces, times, and spheres of life using the conceptual grid of motility.

This paper addresses a interesting and common topic in scientific literature with important practical implications. The objective is relevant and the methodology is adequate. There are, however, some aspects that should be reviewed. In my opinion, would be necessary to resolve some doubts regarding the method and to justify and expand some aspects of the discussion and conclusions.

RECRUITMENT

Could the authors provide more information about how they recruited the participants?

Response 1 (R1): We provided more details in the survey methodology and in particular the modalities for recruiting participants. The recruitment protocol is detailed in section 3.1. Thus, the recruitment of participants is based on 4 steps:

  1. Constitution of a telephone survey database from the addresses of the communes of Veymerange and Coublevie in France,
  2. Telephone recruitment based on a short recruitment questionnaire to identify households corresponding to socio-demographic and spatial criteria,
  3. Sending a letter detailing the objectives of the investigation to the selected persons,
  4. Second telephone call in order to confirm household interest in the survey and to make an appointment for a face-to-face interview.

Were there participants who did not agree to participate when contacted?

R2: The refusal rate is 24%. This aspect has been clarified in section 3.1 of the article.

Did participants get any compensation?

R3: There was no compensation. This aspect has also been clarified in section 3.1.

What information were participants given about the aims of the study when they were encouraged to participate?

R4: A letter was sent to each household meeting the selection criteria following the telephone recruitment questionnaire (see R1). This letter outlined the framework of the survey (doctoral research) and the themes of the face-to-face interview (satisfaction with daily mobility; rhythm of life of families with children). The dimension on rhythms as a main topic particularly facilitated the acceptance of the survey. Indeed, this topic corresponds perfectly to the daily concerns of the families contacted. This aspect was confirmed by particularly rich interviews.  

Was it necessary for the study to be approved by an Ethical Committee?

R5: Yes, the investigation has been approved by the office of the National Commission for Information Technology and Freedom (CNIL in France). The protocol respects the principles of confidentiality and anonymity of the GDPR. This aspect was specified in section 3.1 of the article.

PARTICIPANTS

I think more information about the participants is needed. Could the authors provide more information about the sociodemographic and professional variables of the participants?

R6: Information about the participants is specified about age, household composition and socio-professional categories. The specific status of cross-border worker for groups from the Thionville agglomeration is also discussed, particularly in terms of higher incomes that allow easier access to services such as a household helper or nanny. 

The authors say that these people share similar socio-demographic characteristics, but which are these variables?

R7: Please, see R6.

For example: employment, age of children, income, and educational level, family living in the same neighborhood, suffering from illness (parents or children). Controlling these variables is important because it can affect the results

R8: As shown by the changes in section 3.1. The different criteria of couple biactivity, household composition, age of the couple, age of the children, socio-professional category and location of home and workplace were controlled.  However, the presence of family members in the neighbourhood was omitted in the household selection but was revealed through the interviews. As the analyses show, family supports are an important component of families' strategies for dealing with time pressures. This aspect should be considered in future surveys.

INTERVIEWS

It would be convenient to provide more information about the content of the interview: indicate the structure and examples of items, order in which the themes were presented and indicate whether the authors relied to design it in a previous interview.

R9: The themes of the interview were further detailed in section 3.2. Examples of questions have also been added.

It would be convenient to include the approximate duration of the interviews and the maximum and minimum duration.

R10: The duration of the interviews varied between 60 and 90 minutes. This was also specified in section 3.2 of the article.

The authors say participants are recruited by phone, but it is unclear how the interviews are conducted. Are interviews also conducted by phone? Is it always the same interviewer?

R11: Participants were randomly recruited by telephone using a pre-constituted sampling frame, a recruitment questionnaire and a letter explaining the objectives of the survey. All interviews were conducted by face-to-face in the households' homes. This aspect was specified in section 3.1 of the article. All interviews were conducted by one of the authors, Guillaume Drevon.

The timing of the interview is also unclear. Is a first contact made to explain the objectives of the study and then a day is arranged to carry out the interview?

R12: Please, see R1.

It is indicated that both members of the couple participate. Is one member present while the other answers? That could affect the results.

R13: Both members of the couple were present during the interview. Although the speaking time was relatively balanced between the two members of the couple, this survey configuration is likely to make certain gendered dominance relationships invisible.  This aspect was discussed in the conclusion and limitations of the survey

RESULTS

I find the two approaches (qualitative and quantitative) interesting, but I have doubts about how they were carried out.

Quantitative analysis: Did the authors establish prior criteria on which words or expressions were to enter the analysis of occurrences? What were the sub-themes of the interview? Was there a previous synonym analysis? Only verbs and nouns related to the topic were included? 

R14: The quantitative analysis was carried out based on the structure of the grid of the semi-structured interview. Each topic of the interview was analysed separately from the corpus collected from the families. The lexical fields were not constructed previously, they were revealed from the analysis of the occurrences. This aspect has been clarified in the section 3.2.

Qualitative analysis: How many researchers participated in this analysis? Was an agreement between judges necessary? Was some kind of program like Atlas-ti used?

R15: The analyses were carried out by Guillaume Drevon, who also conducted the interviews. Iramuteq was used as software in order to analyse the semantic corpus.

DISCUSSION

The discussion is well argued, but the authors should deepen further into the implications of the results. The relationship between hypotheses and discussion is also not clearly seen. Do the authors think that the strategies expressed by parents are appropriate? They must deepen more into the solutions or skills that couples must have or about the kind of strategies to improve those skills or resources.

R16: The relationship between hypotheses and discussion has been added in the conclusion. Moreover, we believe that the strategies developed by parents are appropriate, but they highlight important potential inequalities depending on family resources. The case of single-parent families with limited resources is mentioned in the discussion.

The limitations of the study must be included.

R17: The limitations of the study are now integrated into the discussion, notably the size of the sample, which remains limited, the configuration of the interview with both members of the couple and the interest of comparison with more vulnerable families (single parent, low income). 

Finally, the authors must clearly indicate what the study contributes to the literature that already exists.

R18: The contribution of research, particularly with regard to the concept of motility, was developed in the discussion. Indeed, the results make it possible to complete the concept of motility at two levels. First, they make it possible to integrate the skill relating to time management and the reconciliation of places and times of daily activities. They also allow us to put into perspective the importance of social skills for the management of daily life, both in terms of negotiation and in the constitution of a supportive social network. 

In summary, I think the study is interesting, but it is necessary to provide more information on the issues outlined above.

Reviewer 2 Report

I already did this

Author Response

We would like to thank reviewers for their highly valuable comments. The reviews were particularly beneficial for the improvement of the article, especially in terms of clarifying the methodology and the contributions in relation to the current state of knowledge.

Our responses to the reviews are formally presented as follows: referee comments in black and responses in blue. In addition, the main changes have been reported in green font in the article.

As a general comment, please note that the language has been improved with the help of a professional English proofreader.

Reviewer 2

No specific comments

Reviewer 3 Report

This is a very well documented research in the field. 

This paper has an interesting subject in the field and relates outcomes in a well-documented manner. The concept, as well as the methodology and data analysis, are clearly presented. The survey - given its peculiarities- addresses all relevant parameters for car-dependent households residing in sparsely populated areas which are poorly served by PT. There is also adequate analysis of family habits and their skills along with the role of supporting networks. 

A very interesting issue for further research would be to extend the research and compare results for different family types residing in different urban areas (i.e. located in CC compared to suburban) having different transportation options, including soft mobility modes (i.e. walking and cycling).

Moreover, it would be stimulating to relate the mode of transport with the sequence of children's activities and families' agreement in densely populated environments. 

Lastly, it would also be interesting to explore how single-parent families cope with daily rhythms and how global phenomena tend to alter working habits and children's time frames.  

Author Response

We would like to thank reviewers for their highly valuable comments. The reviews were particularly beneficial for the improvement of the article, especially in terms of clarifying the methodology and the contributions in relation to the current state of knowledge.

Our responses to the reviews are formally presented as follows: referee comments in black and responses in blue. In addition, the main changes have been reported in green font in the article.

As a general comment, please note that the language has been improved with the help of a professional English proofreader.

Reviewer 3

This is a very well documented research in the field. 

This paper has an interesting subject in the field and relates outcomes in a well-documented manner. The concept, as well as the methodology and data analysis, are clearly presented. The survey - given its peculiarities- addresses all relevant parameters for car-dependent households residing in sparsely populated areas which are poorly served by PT. There is also adequate analysis of family habits and their skills along with the role of supporting networks. 

A very interesting issue for further research would be to extend the research and compare results for different family types residing in different urban areas (i.e. located in CC compared to suburban) having different transportation options, including soft mobility modes (i.e. walking and cycling).

R19: We thank the reviewer for this comment, wich has been integrated in the discussion.

Moreover, it would be stimulating to relate the mode of transport with the sequence of children's activities and families' agreement in densely populated environments.

R20: Even in dense environments, all the children's activities are carried out by car. This implies a strong socialization amongst car use. This aspect related to car dependency is discussed in the conclusion of the article.

Lastly, it would also be interesting to explore how single-parent families cope with daily rhythms and how global phenomena tend to alter working habits and children's time frames.  

R21: We think indeed that that remark is relevant. The comparison with single parent families would be very interesting. We added this comment in the discussion / conclusion. Specifically, it would make it possible to put into perspective forms of inequality in the management of daily rhythms, particularly in link with economic and social resources.

Round 2

Reviewer 1 Report

think the authors have included modifications that have improved the article. In my opinion, they have also incorporated information that clarifies some doubts in the methodology, results and discussion section.

Although the limitations section could be extended (for example, including the need to control the effect of other variables), it is not essential.